# Role of Mine Tailings in the Spatio-Temporal Distribution of Phosphorus in River Water: The Case of B1 Dam Break in Brumadinho

Teresa Cristina Tarlé Pissarra [1], Renata Cristina Araújo Costa [2], Renato Farias do Valle Junior [3], Maytê Maria Abreu Pires de Melo Silva [3], Adriana Monteiro da Costa [4], Luís Filipe Sanches Fernandes [5], Marília Carvalho de Melo [6], Carlos Alberto Valera [7] and Fernando António Leal Pacheco [8,*]

1. Faculty of Agrarian and Veterinary Sciences (FCAV), State University of São Paulo (UNESP), Via Prof. Paulo Donato Castellane, s/n, Jaboticabal 14884-900, SP, Brazil; teresa.pissarra@unesp.br
2. Centro de Investigação e Tecnologias Agroambientais e Biológicas, University of Trás-os-Montes and Alto Douro (UTAD), Ap. 1013, 5001-801 Vila Real, Portugal; renata.criscosta@gmail.com
3. Geoprocessing Laboratory, Uberaba Campus, t Federal Institute of Triângulo Mineiro (IFTM), Uberaba 38064-790, MG, Brazil; renato@iftm.edu.br (R.F.d.V.J.); mayte@iftm.edu.br (M.M.A.P.d.M.S.)
4. Department of Geography, Institute of Geosciences, Federal University of Minas Gerais, Belo Horizonte 31270-901, MG, Brazil; drimonteiroc@gmail.com
5. CITAB—Centre for the Research and Technology of Agro-Environmental and Biological Sciences, University of Trás-os-Montes and Alto Douro (UTAD), Ap. 1013, 5001-801 Vila Real, Portugal; lfilipe@utad.pt
6. Secretaria de Estado de Meio Ambiente e Desenvolvimento Sustentável, Cidade Administrativa do Estado de Minas Gerais, Rodovia João Paulo II, 4143, Bairro Serra Verde, Belo Horizonte 31630-900, MG, Brazil; marilia.melo@meioambiente.mg.gov.br
7. Coordenadoria Regional das Promotorias de Justiça do Meio Ambiente das Bacias dos Rios Paranaíba e Baixo Rio Grande, Rua Coronel Antônio Rios, 951, Uberaba 38061-150, MG, Brazil; carlosvalera@mpmg.mp.br
8. CQVR—Chemistry Centre of Vila Real, University of Trás-os-Montes and Alto Douro (UTAD), Ap. 1013, 5001-801 Vila Real, Portugal
* Correspondence: fpacheco@utad.pt

**Abstract:** Human actions in the drainage network of hydrographic basins interfere with the functioning of ecosystems, causing negative impacts on the environment. Among these impacts, mass loads with a high concentration of phosphorus (P) have a significant potential for point and diffuse pollution of freshwater. The objective of this work was to model P spatially in the Paraopeba River basin, namely in the main water course and 67 sub-basins, and temporally in the years of 2019, 2020, and 2021, after the rupture of B1 tailings dam of Vale, SA company in Brumadinho (Minas Gerais Brazil). The distribution of total phosphorus concentrations (Pt) in relation to environmental attributes (terrain slope, soil class, and land use) and stream flow was assessed with the help of SWAT, the well-known Soil and Water Assessment Tool, coupled with box-plot and cluster analyses. The Pt were obtained from 33 sampling points monitored on a weekly basis. Mean values varied from 0.02 to 1.1 mg/L and maximum from 0.2 to 15.9 mg/L across the basin. The modeling results exposed an impact on the quality of Paraopeba River water in a stretch extending 8.8–155.3 km from the B1 dam, related with the rupture. In this sector, if the contribution from the rupture could be isolated from the other sources, the average Pt would be 0.1 mg/L. The highest Pt (15.9 mg/L) was directly proportional to the urban area of a sub-basin intersecting the limits of Betim town and Belo Horizonte Metropolitan Region. In general, urban sprawl as well as forest-agriculture and forest-mining conversions showed a close relationship with increased Pt, as did sub-basins with a predominance of argisols and an accentuated slope (>20%). There were various moments presenting Pt above legal thresholds (e.g., >0.15 mg/L), mainly in the rainy season.

**Keywords:** land use change; mining hazards; phosphorus loads; water contamination; watershed management; land use policy

## 1. Introduction

Large river basins are often used for a multiplicity of activities, which makes them vulnerable to many hazards, namely urban-, industrial-, and agriculture-related hazards. In addition, the large dimension turns the basins very complex, with a high heterogeneity of biotic and abiotic elements. This complex interplay between hazard diversity and system heterogeneity leads the large river basins to water insecurity, which is the reason why it is especially necessary to adjust growth and productivity to increase models in these watersheds to sustainability [1–3]. Among other features, sustainable development models must comprise the proposal of land-use scenarios and solutions to contain water pollution and preserve the riverine ecosystems [4,5], framed by water and soil management and conservation programs necessary to lead the development models to success. On the other hand, these programs should be thought to identify the trends of water-polluting agents and how they interact with hydrological processes at the catchment scale [6], since this knowledge is crucial for the implementation of efficient mitigation action where necessary. Depending on the intensity of anthropogenic pressure, these trends relate with climate regulation and the water cycle [7], nutrient cycling and concentration [8], the state of energy transfer in plants [9], and mass movements in the ecosystem change [10].

Phosphorus is among the main indicators of changes to ecosystem processes and water contamination [11–14], because of its strong interaction with the surrounding geospheres and concomitant wide application in activities that frequently impact the rural and urban environments. For example, phosphate fertilization is a widespread soil management practice, operated through the use of phosphorus from fertilizers and animal manure on agricultural land to improve soil fertility and the final crop yields/economic levels [15], but application in excessive amounts results in significant exports to surface and groundwater via runoff and infiltration [16–23]. This chemical element is also present in the superphosphate detergents used on a large scale domestically [24], which may contaminate streams and rivers if not properly removed through wastewater treatment [25–28]. The study of phosphorus dynamics helps to foresee the excess of this nutrient in freshwater [29–31]. Among other standpoints, recent studies have focused on the availability of phosphorus in the stream water network, which depends on the interaction between the compartments that make up the ecosystem [32,33]. They also aimed to understand the processes that determine the entry of phosphorus into the water system and the polluting potential of this element [34]. Some results pointed to the unique feature of phosphorus, which is the low availability in water due to slow diffusion and high fixation in soils [35], or addressed the pathway towards surface water bodies via the erosion process [36–40]. Some other works tackled the mass balance of phosphorus, which is reflected in spatial and temporal distributions of phosphorus concentrations [41,42], and often point to environmental problems such as eutrophication [43–48]. Recent mass balance studies [12,14,34] highlighted the contributions of phosphorus from non-point sources, including agriculture, and also the retention capacity of riparian buffers [49–55]. In large watersheds, however, the assessment of nutrient loads and freshwater concentrations derived therefrom should not focus on a single source. Instead, a comprehensive approach involving multiple sources should be undertaken [32,34,48]. Identifying sources and their relative contributions along drainage networks is an important step in developing management plans to address water quality concerns. This study is an example of that.

The Paraopeba River basin, located in the Minas Gerais state (Brazil), is a large watershed (13,514.94 km$^2$) with significant urban (e.g., Belo Horizonte Metropolitan region), industrial (e.g., mining), and agriculture (e.g., livestock pasturing) occupation. It is therefore adequate for a study on multiple-source modeling of phosphorus concentrations in freshwater. Besides the multiple-use occupation, the Paraopeba basin was affected by the failure of B1 tailings dam, which occurred on 25 January 2019, at the Córrego do Feijão iron ore mine owned by Vale S.A. and located in the municipality of Brumadinho. This catastrophic event released about 11.7 Mm$^3$ of tailings, which spread and buried the bottom of the Ferro-Carvão stream valley up to the confluence with the Paraopeba river nearly

10 km downstream. Besides the most abundant iron and manganese metals, the tailings were composed of phosphorus. About 2.8 $Mm^3$ of these tailings entered the Paraopeba River channel and mixed with the natural sediments. Since then, the tailings continued to be mobilized by the Ferro-Carvão stream towards the Paraopeba River, gradually impacting the fluvial sediments [56]. These materials may have contributed significantly with loads of metals and phosphorus, which are believed to impact the quality of water in the Paraopeba River in the forthcoming years, aggravating the already long-lasting contamination caused by the other sources. The objective of this work was to analyze the spatial and temporal distribution of phosphorus in the drainage network of Paraopeba River basin, in order to understand the role of urban, industrial, and agriculture sources, including the impact caused by the B1 dam rupture. As a specific objective, we aimed to verify if this catastrophic event significantly raised the concentrations of phosphorus in the river water and to what extend along the main channel, as well as if there were differences between the tailings-related contributions of phosphorus measured in the rainy and dry periods. As, to our best knowledge, the assessment of tailings-related phosphorus is lacking in studies of river water quality involving dam breaks, the present work is novel in that regard.

## 2. Materials and Methods

### 2.1. Study Area

The Paraopeba River basin (BHRP) is located in the state of Minas Gerais (Brazil) and the river flows into the southern part of São Francisco River basin (Figure 1). The BHRP extends over a 13,514.94 $km^2$ surface and covers 48 municipalities that shelter approximately 2,349,024 inhabitants.

The climate is tropical and spans two sub-types according to the Köppen's classification: Cwa (hot summer) and Cwb (temperate summer), both with a dry winter. In the basin's lower third, the average temperature is >18 °C irrespective of month, but decreases to values between 15 and 18 °C for at least 1 month in the middle third, and then to 10–15 °C in the upper third. The average annual rainfall is 2285 mm, with a rainy season (October to March) and a dry season (April to September).

The predominant geomorphological characterization is highlighted by the São Francisco Depression, Central-South Mineiro Plateau, East Minas Plateaus, and the Iron Quadrangle [57,58]. The geology is very heterogeneous, with a predominance of itabirites in the Brumadinho region, from which iron and manganese ores are mined. Cover deposits, some itabirite types, and carbonate veins can contain phosphorus hosted in apatite or adsorbed into Mn- and Fe-hydroxides [58,59]. They are considered sterile material and end up deposited in tailings dams.

The main soil units comprise haplic cambisols; red, yellow, and red–yellow latosols; red and red–yellow argisols; and litholic neosols. The cambisols are thin; poorly developed (presence of primary minerals); and, because of their high silt percentage, are low permeability and hence prone to erosion. The latosols are thick and well-drained because the dominant clay and silt particles form very stable and permeable aggregates. They are very old and formed by highly weathered materials, which is the reason why they are poor in nutrients and rich in iron and aluminum oxides (hematite and gibbsite). The argisols are medium-thickness moderate to well-drained soils, characterized by a textural B horizon formed by the accumulation of clay mobilized from the A horizon. The neosols are the thinnest and poorest developed soil types, characterized by mineral or organic A horizon and no B horizon [58].

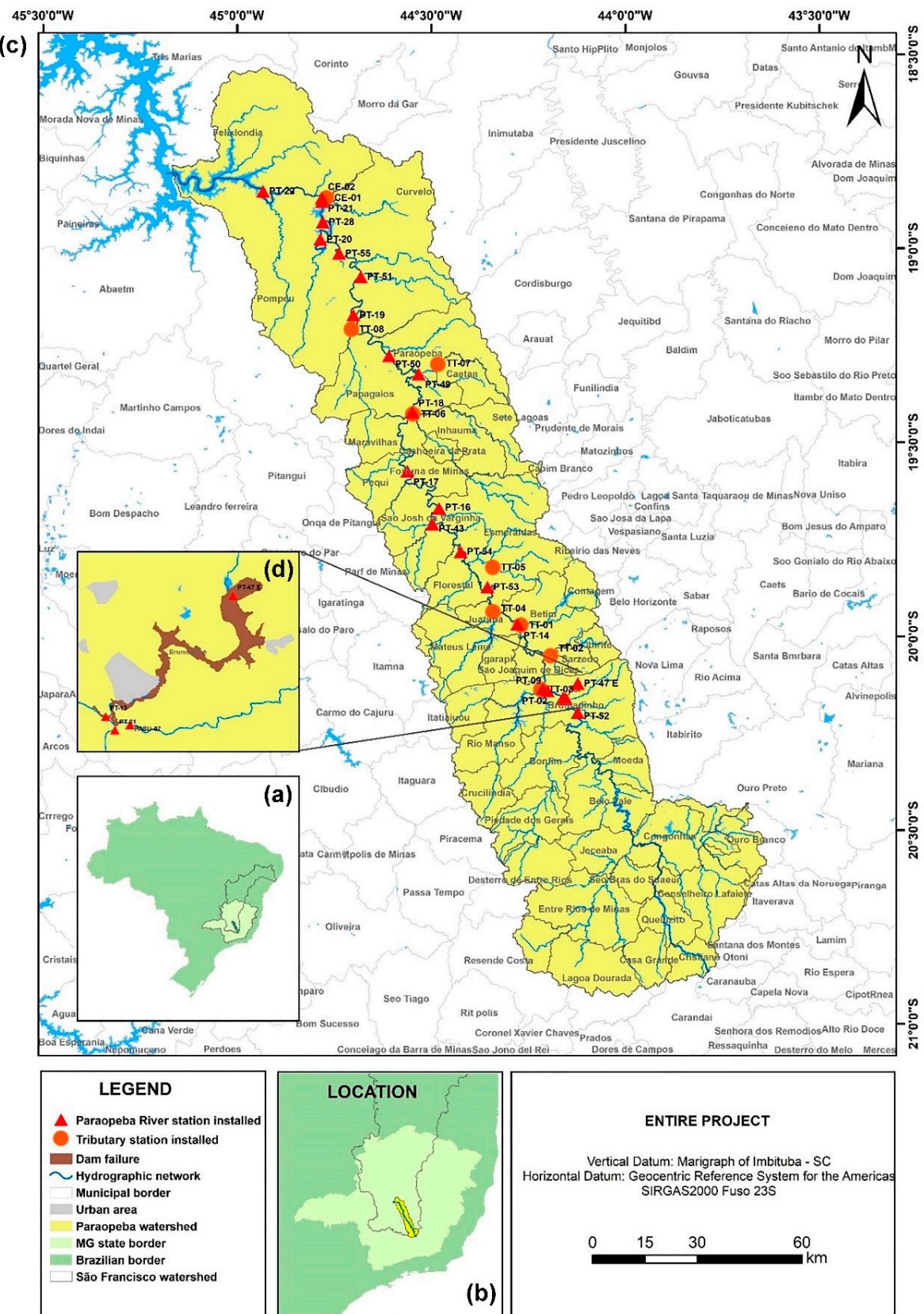

**Figure 1.** Location of Paraopeba River basin in the parent São Francisco River basin (panel (**a**)), Minas Gerais state (**b**) and Brazil (**a**), and distribution of Minas Gerais state municipalities (**c**) within and around the studied region (Ferro-Carvão stream; panel **d**). Identification of monitoring points used in this study (**c**), where phosphorus concentrations were assessed in the 2019–2021 period.

## 2.2. Dataset

The collection of all base maps and temporal records used in this study is summarized in Table 1. The timeframe was mostly the 2019–2021 period, i.e., the 2 years after the rupture of B1 tailings dam. As a first step to assemble the data, the BHRP was divided into compartments using the hydrological model Soil and Water Assessment Tool (SWAT) embedded in the ArcSWAT extension (https://swat.tamu.edu/software/arcswat/, accesses on 26 November 2021). The model details and functioning are described elsewhere [60–63]. The division considered the location of 33 monitoring points used to investigate the spatial and temporal distribution of phosphorus in the basin, which belong to the Vale S.A. company's Emergency Monitoring Plan [64,65]. An automatic procedure resulted in the delimitation 67 catchments, numbered from 1 to 67 (Figure 2). The catchments correspond to a set of hydrologic response units, characterized by homogeneous anthropic action (land use) as well as natural conditions of soil, vegetation cover, and slope. These elements and the meteorologic and hydrologic processes acting on them determine the dynamics of phosphorus concentrations in freshwater in the basin, the understanding of which is the key purpose of this study.

**Table 1.** Database of biogeophysical, chemical, climate, and hydrologic parameters used in the present study.

| Input Data | Format | Image Data 12.5 m Squared Pixels | Period |
|---|---|---|---|
| Digital Elevation Model (DEM) ASF: Alaska Satellite Facility https://search.asf.alaska.edu/, accessed on 26 November 2021 USGS: United States Geological Survey https://earthexplorer.usgs.gov/, accessed on 26 November 2021 | raster | Altitude (meters above sea level) Resampling from USGS | 2011 |
| Land Use MapBiomas: Annual Mapping of Land Cover and Use Project in Brazil http://mapbiomas.org/, accessed on 5 December 2021 | raster | Resampling from Biomes Map vector image | 2019 |
| Soil map FUV: Federal University of Viçosa https://www.dps.ufv.br/?page_id=742, accessed on 5 December 2021 | raster | Resampling from Soil Map vector image | 2010 |
| Climate, accessed on 5 December 2021 CFSR: Climate Forecast System Reanalyzes https://cfs.ncep.noaa.gov/cfsr/, accessed on 12 December 2021 INMET: National Institute of Meteorology http://www.inmet.gov.br/portal/, accessed on 12 December 2021 | raster | Resampling from the raster image of the weather data | 2010 2021 |
| River flow discharge—Minas Gerais Water Management Institute and Vale S.A. company http://www.igam.mg.gov.br/transparencia/dados-abertos, accessed on 12 December 2021 | raster | Resampling from the raster image of the flow data | 2019 2021 |
| Phosphorus Concentration Vale S.A. company (Emergency Monitoring Plan) | | Resampling from the raster image of the phosphorus data | 2019 2021 |

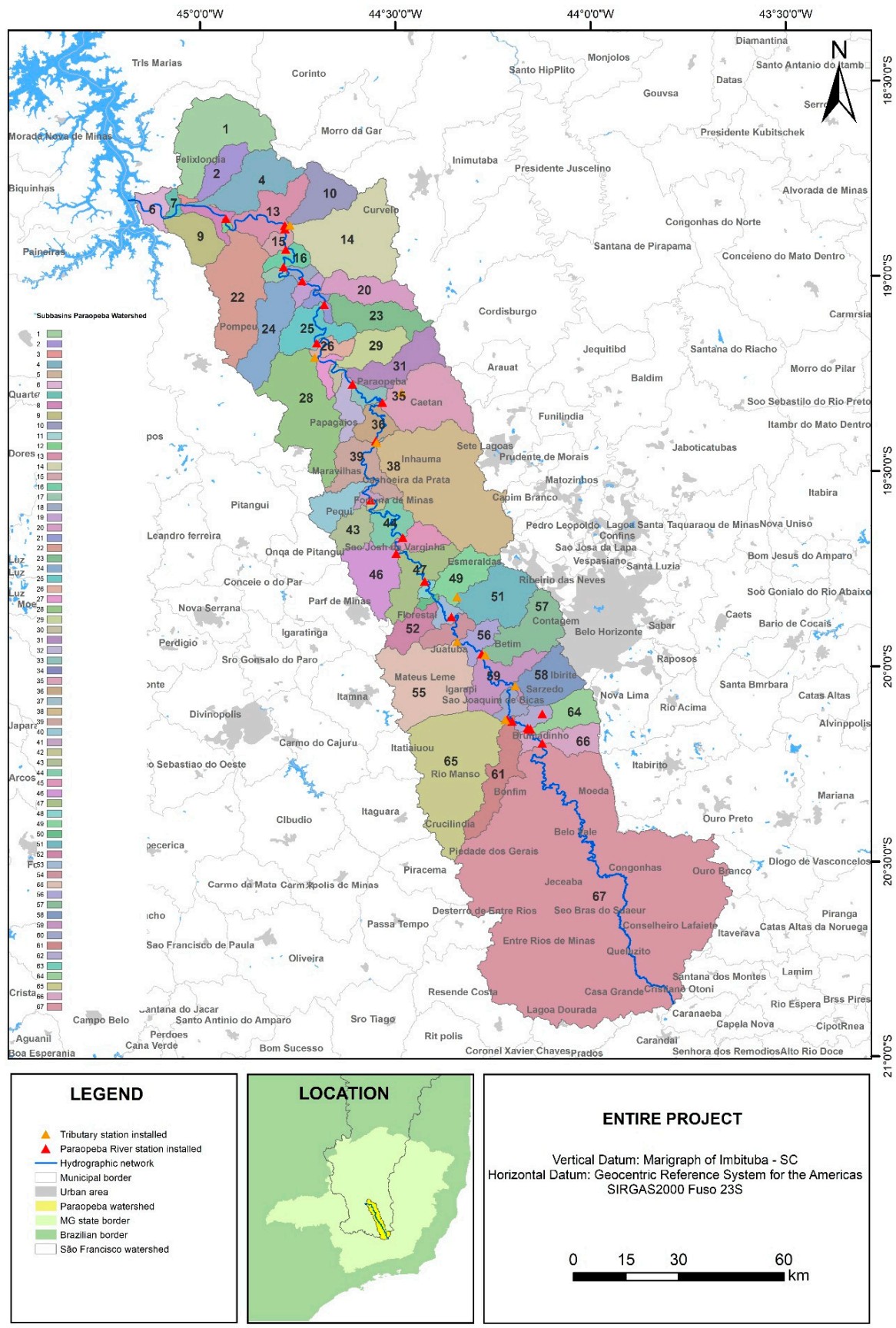

**Figure 2.** Distribution of sub-basins (labeled polygons) within the Paraopeba River basin, as delineated by the SWAT model.

The spatial assessment of the terrain slope was based on a Digital Elevation Model (DEM) covering the entire Paraopeba River basin. The slope classes were set according to the Brazilian Soil Classification System [66]: Plan for slopes from 0 to 3%; smoothly undulated from 3 to 8%; undulated from 8 to 20%; strongly undulated from 20% to 45%; mountainous from 45 to 75%; and scarp for slopes greater than 75%. The soil map classes were also delineated in keeping with the Brazilian Soil Classification System [66]. The land use map was prepared considering the criterion of predominant vegetation cover, as defined in the technical manual of Brazilian vegetation [67], and it also resorted to the Brazilian Annual Land Use and Land Cover Mapping Project (https://mapbiomas.org/, accessed on 5 December 2021). Stream flow data were obtained from historical series of Vale S.A. company and the IGAM's (Minas Gerais Water Management Institute) hydrometric stations covering the 2019–2021 period. Total phosphorus (Pt) concentrations were obtained from chemical analyses of stream water collected at the 33 emergency plan monitoring points located in the main water course (PT sites in Figure 1) and some tributaries (TT sites) of the Paraopeba River, during the years of 2019, 2020, and 2021. The water samples for the analysis of Pt were collected on a daily to weekly basis by the Vale, S.A. company and IGAM, as described in the reports of Arcadis [58,68]. In this study, the dataset was recast to the weekly basis (Supplementary Materials). All the analyses were based on standard methods for the examination of water and wastewater [69]. The stream flow discharges ($m^3/s$) estimated at the 33 monitoring points were multiplied by Pt concentrations (mg Pt/L) determined at the same points to obtain Pt fluxes within the corresponding catchments, expressed as unit of mass per time (ton/year). The procedure was replicated for average and maximum concentrations as well as for the rainy and dry seasons. The constant state condition was assumed according to the guidelines of Gao et al. [41].

*2.3. Methods*

A descriptive statistical analysis was performed in STATISTICA software (https://www.statistica.com/en/ (accessed on 24 April 2022)) to assess the central tendency, spread, and trends in the Pt concentrations. The analysis was replicated for each monitoring station indicated in Figure 1 (panel c) to verify if the concentrations varied longitudinally along the main water course and, if yes, seek for a reason, namely the impact of B1 dam rupture. The results were visualized in the box-plots drawn for each station and built on the references of minimum and maximum values, first and third quartiles, median, and outliers of Pt. The visualization in box-plots meant to detect seasonal variations in the Pt concentrations and check if the rupture of B1 tailings dam impacted the river water quality differently in the rainy and dry seasons, namely in the following periods: Rainy 2019—January to March 2019, right after the B1 dam rupture; Dry 2019—April to September 2019; Rainy 2020—October 2019 to March 2020; Dry 2020—April to September 2020; and Rainy 2021—October 2020 to March 2021.

A multivariate analysis was also performed in the computer program STATISTICA to assess the joint variations of slope percentage, soil class, land use, stream flow, and Pt concentration in each sub-basin, following the cluster method [70]. The hierarchical grouping method was used to assemble the physically homogeneous areas and the Ward's method to maximize their homogeneity based on the values of Euclidean distance. The digital processing of Pt concentration and stream flow maps to generate Pt flux maps resorted to the Map algebra tools of ArcMap (https://www.esri.com/en-us/arcgis/products (accessed on 24 April 2022)) and was replicated for the aforementioned periods.

The evaluation of Pt concentrations from an environmental standpoint was based on the Joint Normative Deliberation COPAM/CERH-MG no. 01, of 5 May 2008. In that context, the measured Pt concentrations were compared to the following maximum admissible values: class 1 freshwater—0.1 mg/L in lotic environment and tributaries of intermediate environments; class 2 freshwater—0.050 mg/L in intermediate environments, with residence time between 2 and 40 days, and direct tributaries of a lentic environment; class 3 freshwater—0.15 mg/L in lotic environment and tributaries of intermediate

environments. Whenever the Pt concentrations were above 0.15 mg/L, the environment was considered to have reached the maximum admissible value of any freshwater class mentioned in the Joint Normative Deliberation COPAM/CERH-MG No. 01, of 5 May 2008.

## 3. Results and Discussion

The spatial distributions of terrain elevation and slope, soil class, and land use or occupation are illustrated in Figure 3a–d. The highest altitudes are located in the southern part of Paraopeba River basin, ranging from 906.3 m to 1619.0 m, while in the northern sector, near the confluence of Paraopeba and São Francisco rivers, the predominant elevation class is from 556 to 685 m (Figure 3a). The slope gradients show a significant spatial variation, with a concentration of smooth classes (0–3% and 3–8%) to the north of the basin, and of steep classes (above 20%) to the south (Figure 3b). This likely causes a differentiation of runoff across the basin and the probability of intense erosion and silting processes in some regions. The soil map (Figure 3c) reveals a predominance of latosols, argisols, cambisols, and neosols. The most weathered soils in the Paraopeba River basin (latosols), considering their contents of iron and aluminum oxides, tend to drain phosphorus into the water compartment, in contrast with soils that contain lower levels of these metals. Thus, the latosols of Paraopeba River basin likely present reduced adsorption capacity and availability of phosphorus for plants, but a high vulnerability to leaching processes. However, it is important to note that if phosphorus is present in the labile form resulting from fertilization, it may be available in larger amounts in the system. In general, the soil overlying a catchment coupled with its weathering stage will indicate if the catchment represents a source or sink of phosphorus. The map of land use and land cover reproduced from the Mapabiomas 2019 database (Figure 3d) reveals a predominance of mosaic areas of agricultural occupations in forest areas and managed pastures, totaling 60.6% of Paraopeba River basin. Forest areas represent 4.4%, while reforestation (silviculture) occupies an area of 4.79%. Urban centers correspond to 3.46% of Paraopeba River basin, being mostly located on the slopes of some catchments.

### 3.1. Spatial and Temporal Distribution of Total Phosphorus Concentrations

The spatial distribution of total phosphorus (Pt) concentrations is illustrated in Figure 4. The figure shows an order of magnitude difference in the Pt concentrations across the Paraopeba River basin, either in the average Pt that varied from 0.02 to 1.1 mg/L (Figure 4a) or the maximum Pt that ranged from 0.2 to 15.9 mg/L (Figure 4b), during the 2019–2021 period. The sub-basins most affected by average Pt concentrations clearly above the largest threshold of Joint Normative Deliberation COPAM/CERH-MG No. 01 (0.15 mg/L) were catchments no. 31 and 35 in the northern part, and catchments no. 51, 52, 54, 55, 56, 57, 58, 59, 60, and 65 in the southern part of the basin. As regards maximum Pt concentrations, the most affected sub-basins were the no. 35 and 57. Sub-basins 56 and 57 are located downstream of B1 dam site, meaning that the mud wave derived from the dam rupture may have caused these hotspots. It is worth noting, however, that artificial urban areas can also cause a great anthropogenic pressure on the water courses, and therefore may have also contributed to the largest average (1.1 mg/L) and maximum (15.9 mg/L) Pt values observed in catchment no. 57, due to the concentration of population and industries (cf. land use in Figure 3d).

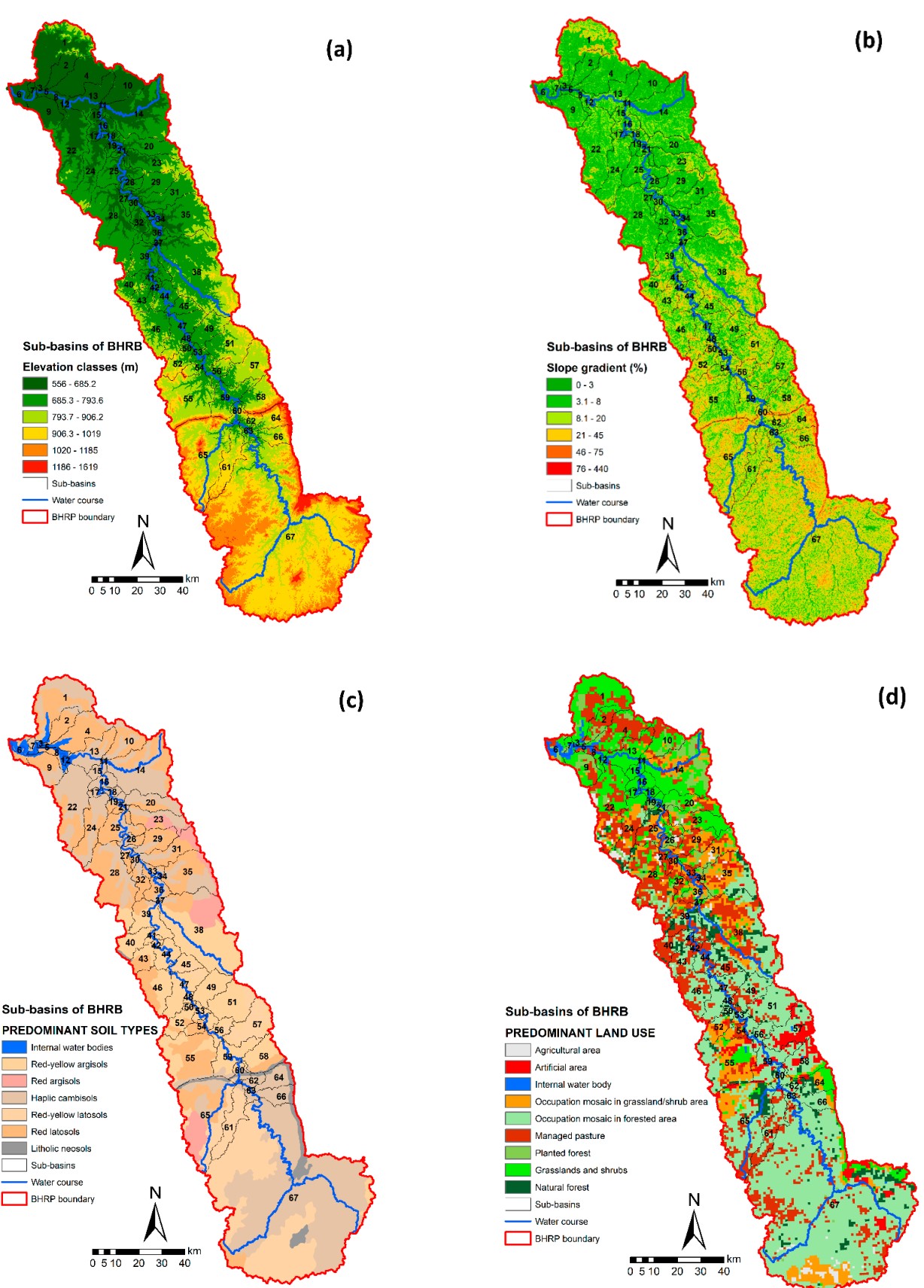

**Figure 3.** Maps of altitude (**a**), hill slope (**b**), dominant soil type (**c**), and land use (**d**) in the BHRP.

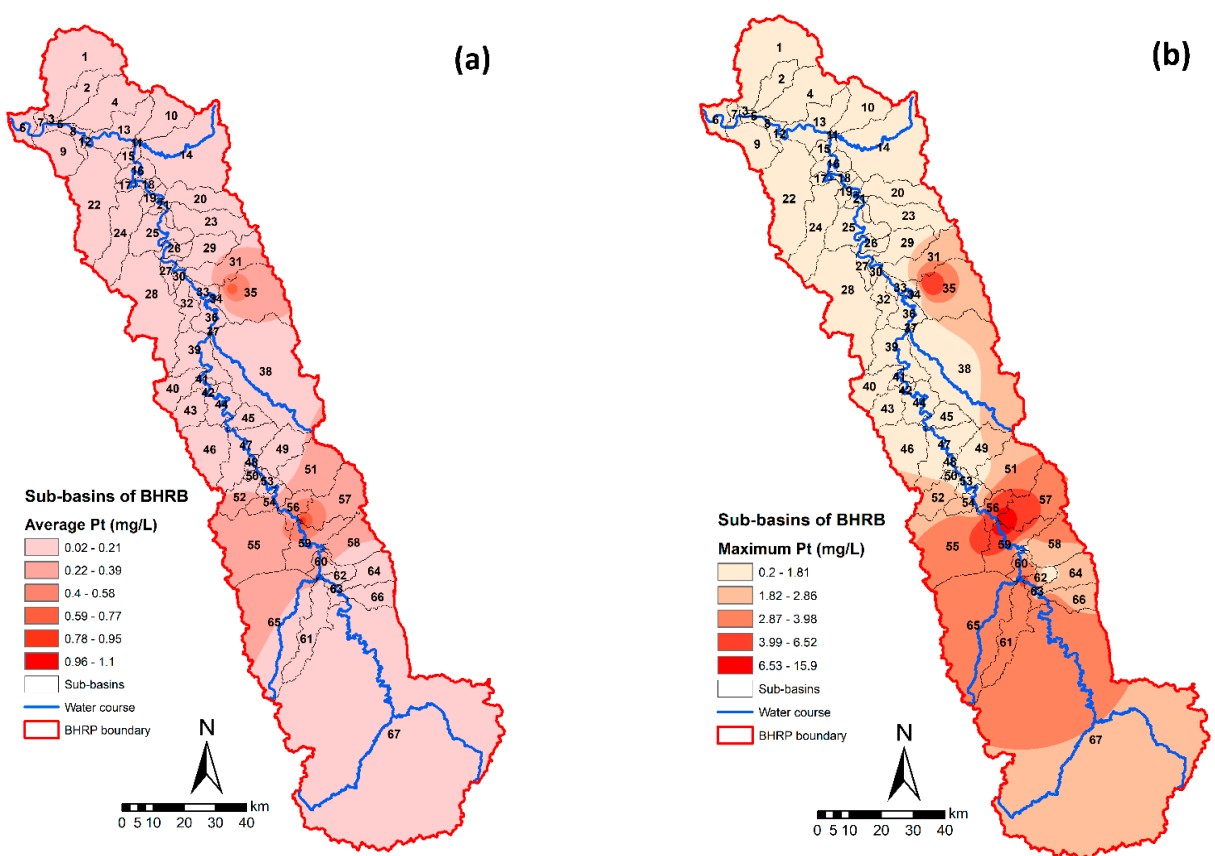

**Figure 4.** Mean (**a**) and maximum (**b**) concentrations of total phosphorus (Pt; mg/L) across the BHRP.

Figure 5 represents the minimum, maximum, first and third quartiles, median, and outliers of Pt concentrations in the monitoring stations located upstream of B1 dam site, discriminating the rainy and dry seasons of period 2019–2021. The stations represented in the diagram are PT-52, PT-01 (in the Paraopeba River), and PT-47E (in the Ferro-Carvão stream), and receive drainage from sub-basins no. 62, 66, and 67. The box-plot of PT-52 points to an average Pt concentration of 0.07 mg/L, taking the seasons altogether, which is below the legal threshold defined for class 2 freshwater (0.1 mg/L), while for station PT-01 the average concentration was 0.11 mg/L, which is below the legal threshold defined for class 3 freshwater (0.15 mg/L). The diagram also highlights various outliers, especially the ones observed in the rainy season of 2019 at PT-01 and rainy season of 2020 at PT-52, where the Pt concentrations reached 3.18 mg/L and 1.81 mg/L, respectively. The region of sub-basins no. 66 and 67 is characterized by incipient sewage treatment [71], and areas where agriculture and livestock production are practiced (Figure 3d), with potential wash out of phosphorus fertilizers in runoff and transport towards PT-01 and PT-52, especially in the periods of intense precipitation (rainy seasons). At these stations, the sporadic very high concentrations of total phosphorus could not be related with the rupture of B1 dam, because of their location upstream of this site. In these sub-basins, prevention of anomalous Pt values could be accomplished with implementation of good management practices for conservation of soil and water, which would minimize the impacts of untreated sewage and leaches of fertilizers. Station PT-47E is the closest to the area of B1 tailings dam, but the available data is restricted to 2020. Thus, for this station located in the sub-basin no. 62, it was not possible to analyze the effect of B1 dam rupture that occurred in January 2019. Figure 6 displays box-plots similar to Figure 5, but relative to three stations located downstream and relatively close to the Ferro-Carvão stream–Paraopeba River confluence, namely stations PT-13, PT-09, and PT-02. In these stations, the rainy season of 2019 (immediately after the dam rupture) is clearly different from the other seasons,

because of its number and magnitude of outliers that reached maximum Pt concentrations of 3.41, 5.25, and 5.11 mg/L, respectively, and average concentrations between 0.14 and 0.18 mg/L. In this case, the contribution of B1 tailings dam rupture to the anomalous Pt concentration seems unquestionable.

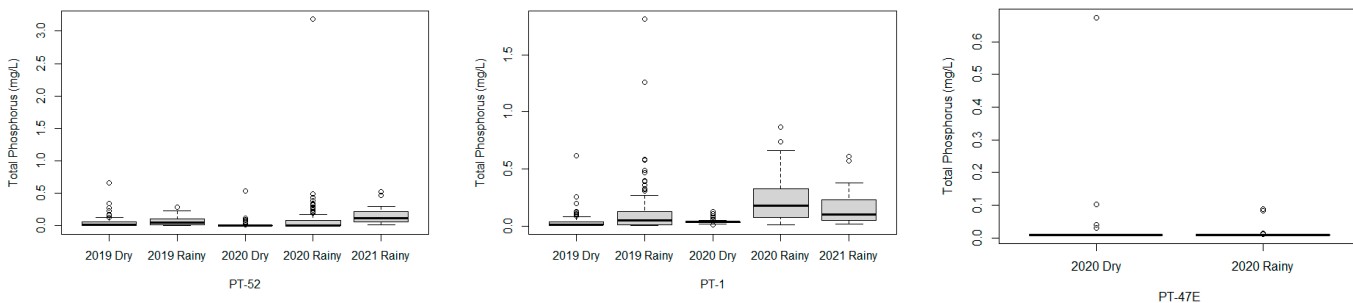

**Figure 5.** Box-plots of total phosphorus concentrations in stations located upstream of B1 tailings dam, with discrimination of season. Station PT-52 received drainage from sub-basin no. 67, station PT-01 from sub-basins no. 66 and 67, and station PT-47E from sub-basin no. 62. The sub-basins are indicated in Figure 1.

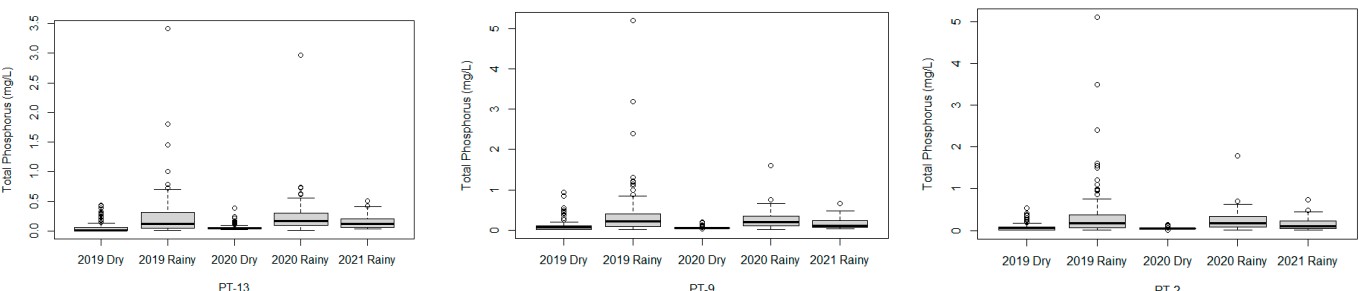

**Figure 6.** Box-plots of total phosphorus concentrations in stations located downstream of B1 tailings dam, with discrimination of season. The distances of these stations from the B1 dam site are: 8.8 km (PT-13), 14.1 km (PT-09), and 18.7 km (PT-02).

The results relative to stations PT-13, PT-09, and PT-02 in the rainy season of 2019 are worth an additional comment. The presence of phosphorus in the tailings is firstly justified by its presence in the mined itabirites and associated carbonate veins, in the form of apatite or adsorbed in Fe and Mn oxyhydroxides [59]. Since phosphorus is an ore contaminant, its removal occurs during the ore-processing phase through physical, (bio)chemical or thermal treatments, and the P-rich residue ends up in the tailings [72]. The tailings released from the B1 dam after the rupture formed an 8.8 km-long deposit over the Ferro-Carvão stream that further extended along the Paraopeba River. On its path, the mud wave composed of tailings eroded and mixed with soils from the Ferro-Carvão stream margins, and then this material mixed further with the Paraopeba River natural sediments. Before the rupture, the margins of Ferro-Carvão stream were mostly used for agriculture, and hence the eroded soils likely accumulated large amounts of nutrients over the years, including phosphorus. Thus, apart from the contribution of phosphorus contained in the tailings, the anomalous Pt concentration observed in the rainy season of 2019 may also reflect the contribution of eroded soils.

The distribution of Pt concentrations in Paraopeba River tributaries located less than 55 km from the B1 dam site, namely stations TT-03 (19.2 km), TT-02 (30.3 km), and TT-01 (54.2 km), differed significantly from the distributions of previously described stations located in the main watercourse. The above-mentioned tributary stations are located at the outlets of sub-basins no. 65, 58, and 57, respectively, and the seasonal distributions of Pt concentrations are illustrated in Figure 7. The average Pt concentrations were 0.27 mg/L

in TT-03 (range: 0.004–1.21 mg/L), 0.32 mg/L in TT-02 (range: 0.004–1.65 mg/L), and 1.14 mg/L in TT-01 (range: 0.004–15.9 mg/L). Thus, the box-plots in this figure reveal a smaller number of outliers but higher average concentrations when compared with those represented in Figures 5 and 6. A possible cause for the general higher concentrations observed in the tributaries could be less dilution, because the tributary streams discharge much smaller volumes of water when compared to the Paraopeba River. However, the very large value observed in the TT-01 (1.14 mg/L, on average) likely has the contribution of other effects besides dilution. In this case, the predominance of urban areas in the drainage basin (catchment 57; cf. land use in Figure 3d), namely the urban areas of Betim municipality and surrounding areas of Belo Horizonte metropolitan region, may be the additional (and principal) effect. In the last decades of the 20th century, there was a considerable increase of municipal urban areas, including of Betim's and Belo Horizonte. Usually, phosphorus pollution in the urban environment results from untreated discharge of domestic sewage and waste waters that contain P-bearing detergents [24,38–41].

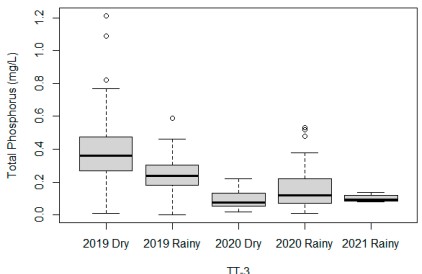 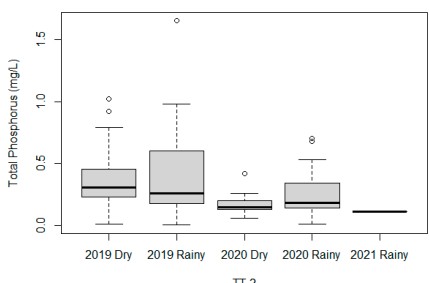 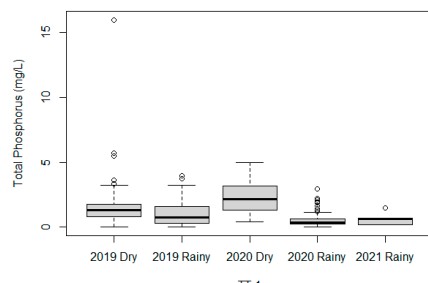

**Figure 7.** Box-plots of total phosphorus concentrations in stations located at the outlets of Paraopeba River tributaries: TT-03 (19.2 km from the B1 dam site), TT-02 (30.3 km), and TT-01 (54.2 km).

The seasonal distributions of Pt concentrations in stations of Paraopeba River and tributaries located farther than 55 km from the B1 dam site are depicted in Figure 8. As in former stations, the largest values mostly occurred in the rainy season, an outcome that reinforces the presumed relationship of high Pt concentrations with wash-out processes of soils during the periods of intense precipitation.

In order to understand Pt concentrations across the Paraopeba River basin without the influence of tributaries, and hence highlight the role of B1 dam rupture, we deleted from Figure 4a the concentrations of TT stations and redraw that figure as Figure 9. As a result, a sector of higher Pt concentrations is highlighted between the Ferro-Carvão stream–Paraopeba River confluence and station PT-17 located 155.3 km away from the B1 dam site. Considering the beginning of this sector immediately after that junction and the contrast of Pt concentrations in the sector and outside, the stretch of Paraopeba River located between PT-13 and PT-17 (8.8–155.3 km far from the B1 tailings dam) can be considered influenced by the dam rupture in the 2019–2021 period. This sector spans the so-called "anomalous" and "transition" sectors defined by Pacheco et al. [56] based on Fe/Al ratios measured in stream sediments.

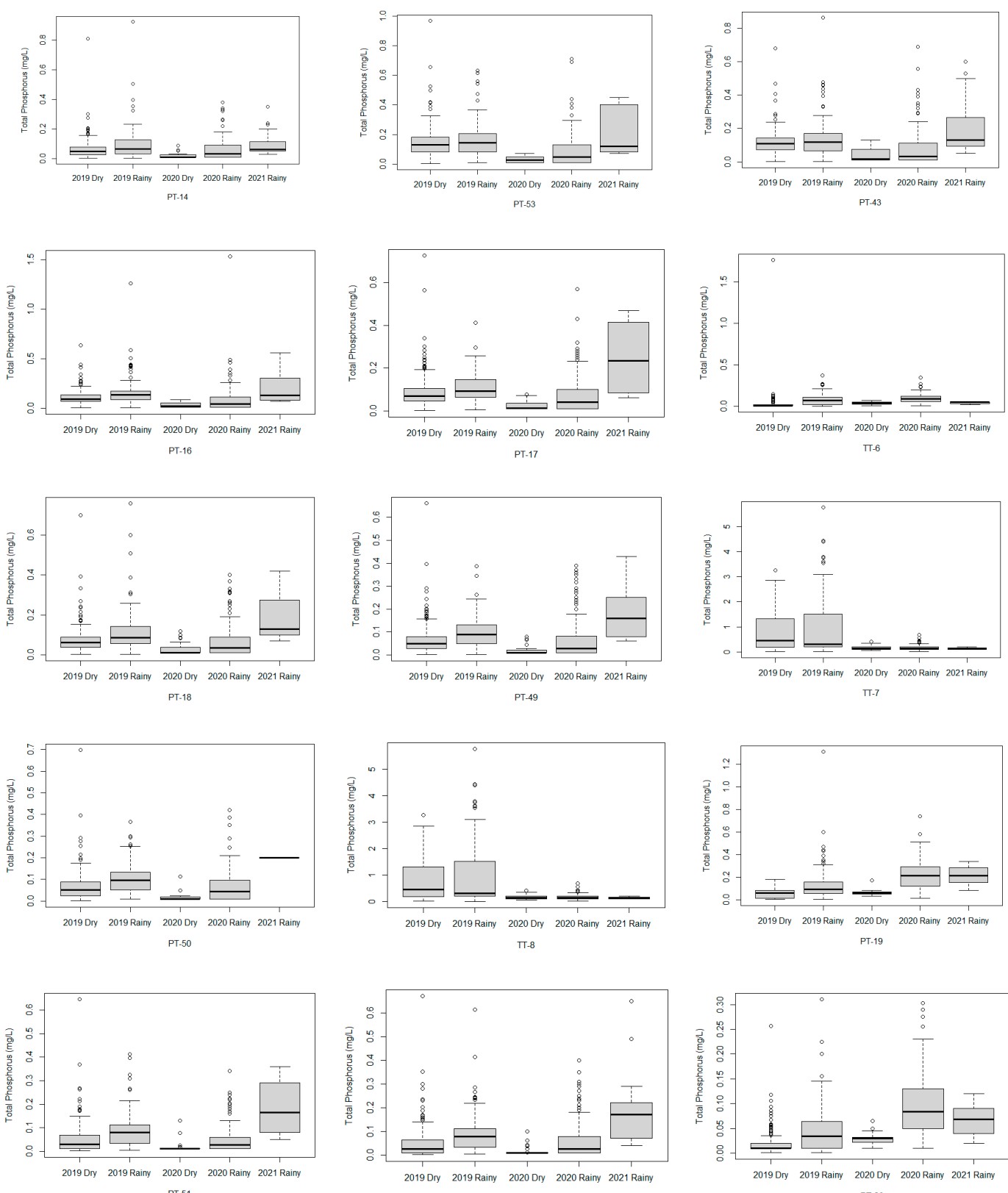

**Figure 8.** *Cont.*

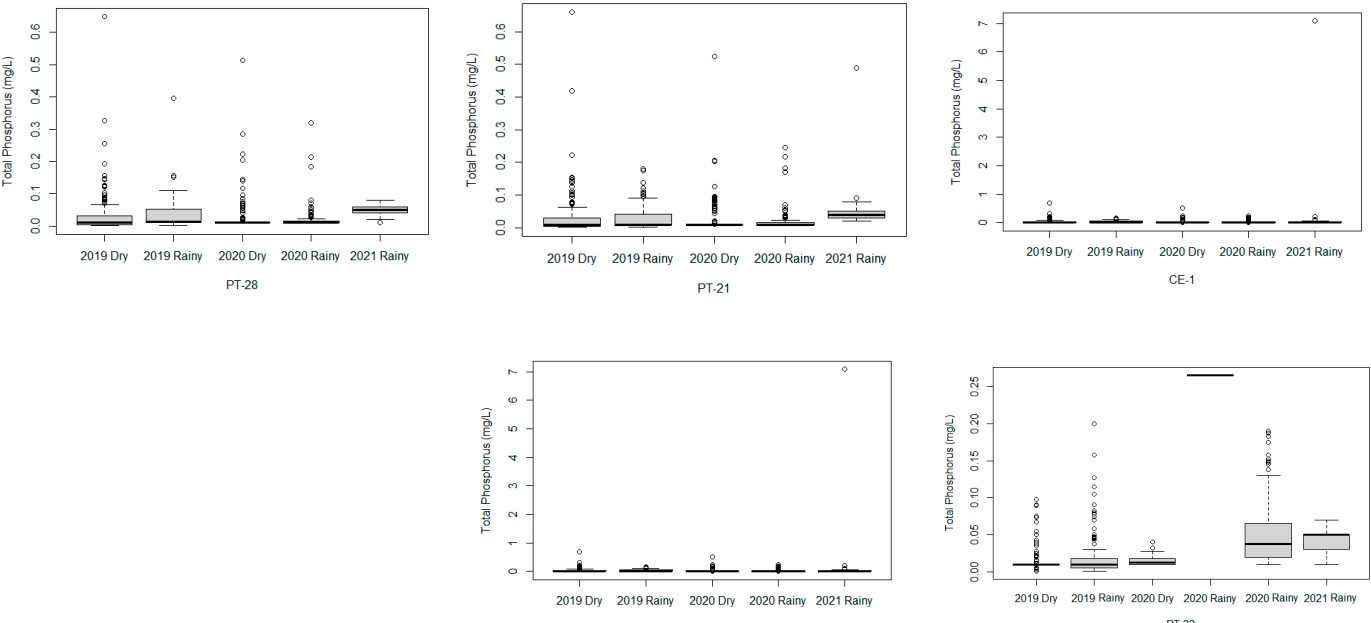

**Figure 8.** Box-plots of total phosphorus concentrations in stations located in the Paraopeba River (PT stations) or in the tributary streams (TT stations) farther than 55 km from the B1 tailings dam site: PT-14 (53.9 km); PT-53 (76.5 km); PT-43 (115.8 km); PT-16 (122.6 km); PT-17 (155.3 km); TT-06 (183.7 km); PT-18 (184.2 km); PT-49 (206.3 km); TT-07 (206.3 km); PT-50 (222 km); TT-08 (239.5 km); PT-19 (249.8 km); PT-51 (267 km); PT-55 (282 km); PT-20 (297.9 km); PT-28 (309 km); PT-21 (315 km); CE-01 (315.7 km); CE-02 (319.3 km); and PT-22 (341.6 km).

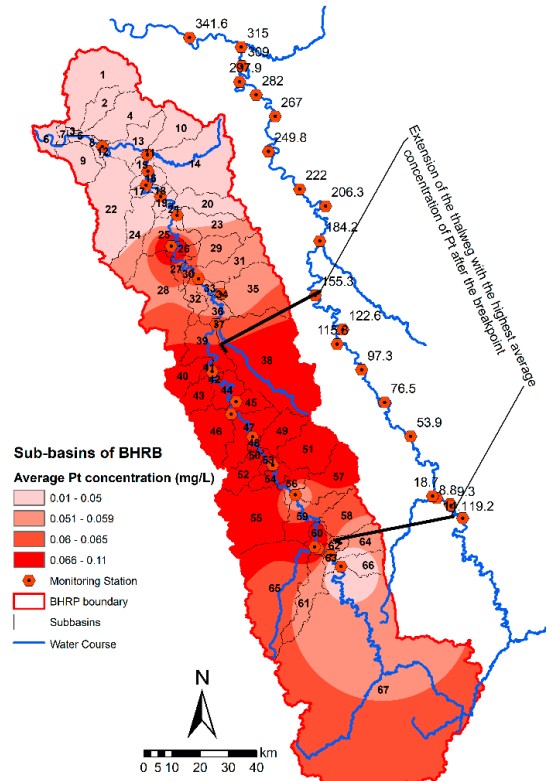

**Figure 9.** Average concentrations of total phosphorus (Pt; mg/L) across the BHRP, with the identification of a sector located between station PT-13 (8.8 far from the B1 dam site) and PT-17 (155.3 km), considered as influenced by the B1dam rupture during the period 2019–2021.

### 3.2. Relationship between Pt Concentrations and Environmental Settings

The joint analysis of Pt concentrations (minimum, mean, and maximum) measured at the monitoring stations (Figures 1 and 2) and land uses assessed within the corresponding drainage areas (Figure 3d) is represented in the dendrogram of Figure 10. The concentration of phosphorus is primarily connected with the urban occupation (linkage distance 6–7), emphasizing the importance of domestic effluents as a key source of phosphorus pollution in the Paraopeba River basin. The cluster of Pt concentrations and urban areas connect with the forest occupation at a linkage distance of 8–9, an outcome that is interpreted as deforestation associated primarily with the urban expansion. At a similar but larger distance, this later cluster connects with agriculture and mining areas, meaning that forest-agriculture and forest-ore mining conversions are also important active land use changes in the basin. Considering the larger distances representing the connection of Pt concentrations with the expansion of mining, the phosphorus pollution derived from this activity has to be considered less impacting than the contamination caused by the urban sprawl. Strikingly, activities such as forestry or livestock pasturing seem to play minor roles in the distribution of Pt concentrations in the Paraopeba River basin. The role of various environmental settings on the spatial patterns of phosphorus, including of land use, has been broadly addressed in a diversity of works worldwide [13,14,32,34,48].

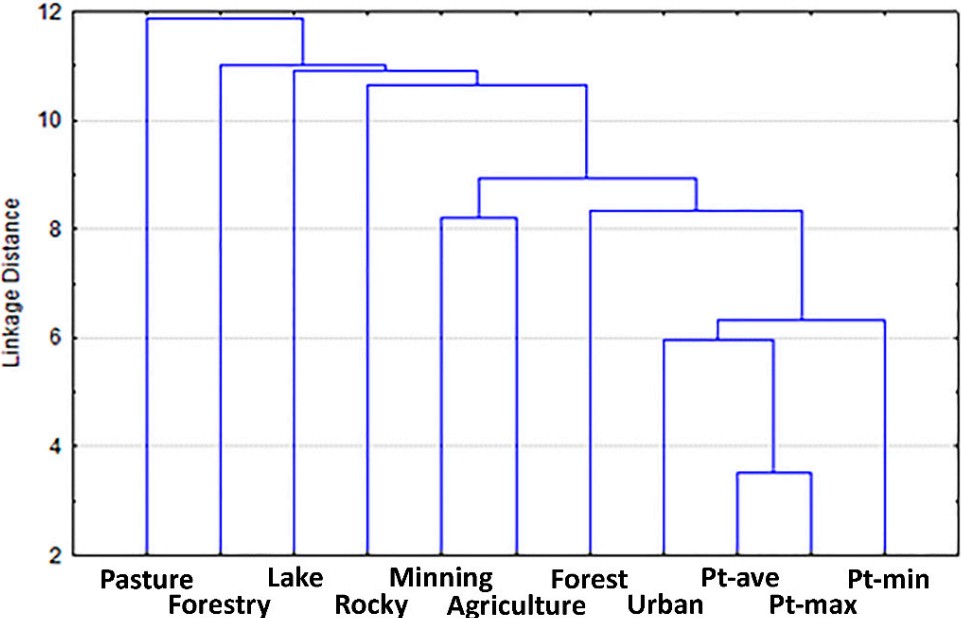

**Figure 10.** Joint analysis of Pt concentrations and land uses based on a hierarchical cluster analysis applied to the 67 sub-basins. Symbols: pasture, forestry, lake, rocky, mining, agriculture, forest, and urban, are land uses or occupations; Pt_ave, Pt_max, and Pt_min are average, maximum, and minimum total concentrations of phosphorus in water.

The clustering of Pt concentrations with soil types and terrain slope is portrayed in Figure 11. The results suggest a close relationship between large Pt concentrations, steep slopes (>20%), and argisols. On the other hand, flat to undulating landscapes (slopes < 8%) and latosols seem to be much less related with the Pt concentrations in the water of Paraopeba River and tributaries. The results concerning the argisols are not surprising, because these soils comprise a textural B horizon that, when associated with a location in steep hillslopes, increases their susceptibility to erosive processes and, consequently, causes the dragging of particles that will be deposited along the watercourse and later may desorb phosphorus into the water column [38]. Therefore, sub-basins with a terrain slope above 20% in the Paraopeba River basin should be priority areas for conservation management and immediate action capable to minimize the impact of human activities.

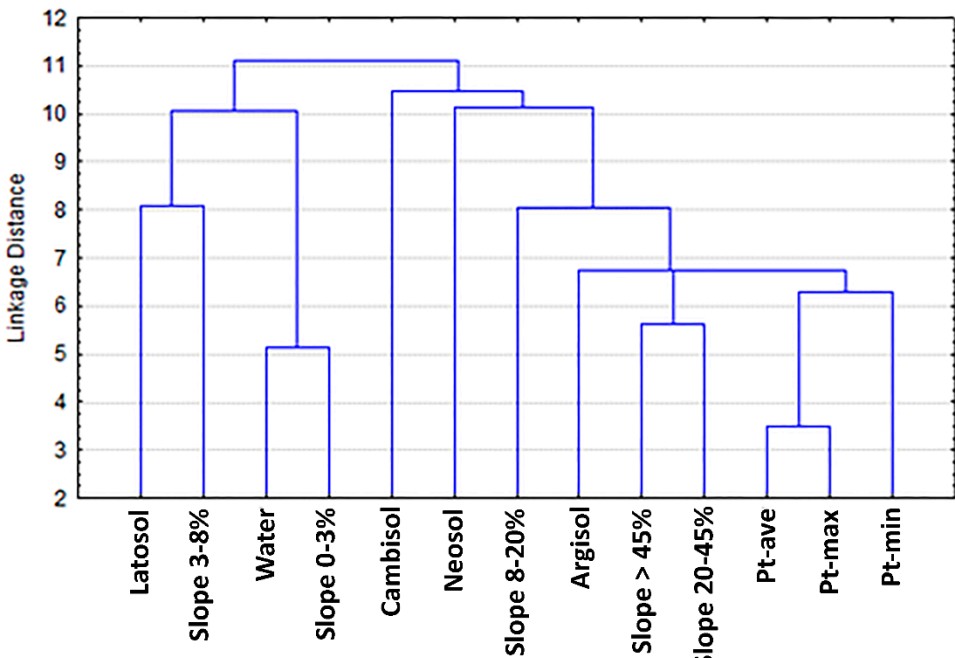

**Figure 11.** Joint analysis of Pt concentrations, terrain slope, and soil types, based on a hierarchical cluster analysis applied to the 67 sub-basins. Symbols: latosol, cambisol, neosol, and argisol, are soil types as indicated in Figure 3c; the slope classes are in keeping with the slope map of Figure 3b; Pt-ave, Pt-max, and Pt-min are average, maximum, and minimum total concentrations of phosphorus in water.

The different contributions of soil classes to the concentration of total phosphorus in the waters of Paraopeba River and tributaries can also be related to intrinsic soil properties. More weathered soils such as the latosols, which present a balance of electropositive charge due to the presence of Fe and Al oxides and hydroxides, tend to behave as a phosphorus sink (low release to water). In these soils, phosphorus is adsorbed to clays and therefore is not labile under normal environmental conditions [39,40]. However, a decrease of soil redox potential (Eh) may trigger the desorption of phosphorus from the host and, consequently, its release into the aquatic medium [73]. Considering the large volume of soil washed away from the Córrego do Feijão basin after the rupture of B1 tailings dam, which was potentially fixing phosphorus, it is likely that there was a release of large amounts of this element into the aquatic medium, because the redox potential of soil in situ is different from soil deposited on a riverbed and is mixed with ore tailings and sediments.

### 3.3. Relationship of Pt Concentration and Stream Discharge and Fluxes of Pt

The long-term average stream discharge in the Paraopeba River basin is illustrated in Figure 12 for the rainy (Figure 12a) and dry (Figure 12b) seasons of the 2019–2021 period. In the first case, the values varied from 17.86 $m^3$/s to 213.7 $m^3$/s, and in the second case from 9.45 $m^3$/s to 98.54 $m^3$/s. When these data are correlated with the Pt concentrations through the Spearman rank-order method, significant negative coefficients occur between the average Pt and the average and maximum stream discharge (Qave and Qmax in Table 2), in both the rainy and dry seasons. This is strong evidence of a significant dilution effect, which is less evident during the period of stream flow recession (for $Qr_{min}$, the coefficients are negative but not significant at $p < 0.05$).

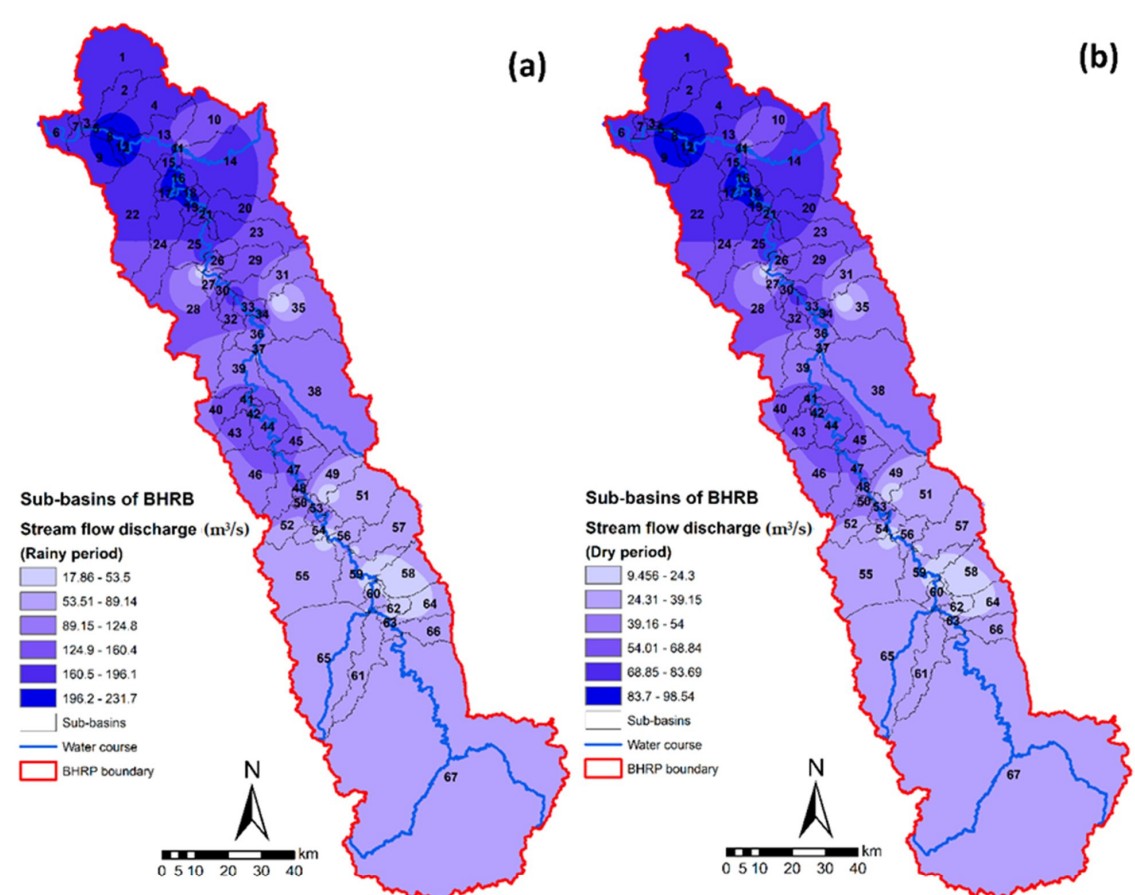

**Figure 12.** Stream discharge in the Paraopeba River basin: (**a**) in the rainy period; and (**b**) in the dry season of 2019–2021 period.

**Table 2.** Spearman rank-order correlation coefficients between total phosphorus concentrations (Pt; mg/L) and minimum (subscript min), maximum (max), and average (ave) stream discharge in the Paraopeba River basin in the rainy (Qr; $m^3/s$) and dry (Qd; $m^3/s$) seasons of 2019–2021 period. The coefficients highlighted in boldface are significant for $p < 0.05$.

| Parameter | $Pt_{min}$ | $Pt_{max}$ | $Pt_{ave}$ |
|---|---|---|---|
| $Pt_{min}$ | 1 | | |
| $Pt_{max}$ | 0.50 | 1 | |
| $Pt_{ave}$ | 0.70 | **0.91** | 1 |
| $Qr_{min}$ | −0.37 | −0.50 | −0.63 |
| $Qr_{max}$ | −0.69 | −0.47 | **−0.71** |
| $Qr_{ave}$ | −0.61 | −0.57 | **−0.78** |
| $Qd_{min}$ | −0.37 | −0.50 | −0.63 |
| $Qd_{max}$ | −0.69 | −0.47 | **−0.71** |
| $Qd_{ave}$ | −0.61 | −0.57 | **−0.78** |

The correlation results presented for the dry period recommend that phosphorus monitoring in some sub-basins should be rigorous, considering that the concentration of this element increases in relation to the contribution by the use and management of the soil and the lower volume of water. The importance of implementing good management practices for the control of phosphorus pollution in the Paraopeba River basin is highlighted by the spatial distribution of Pt fluxes (t/year) obtained from the product of Pt concentration (mg/L) and stream discharge ($m^3/s$). In the 2019–2021 period, the fluxes varied between 130.87 and 811.82 t/year in the rainy season and from 56.47 to 361.7 t/year in the dry season

(Figure 13a,b). These figures confirm the anomalous sector already reported for the Pt concentrations and related with the impact of B1 dam rupture.

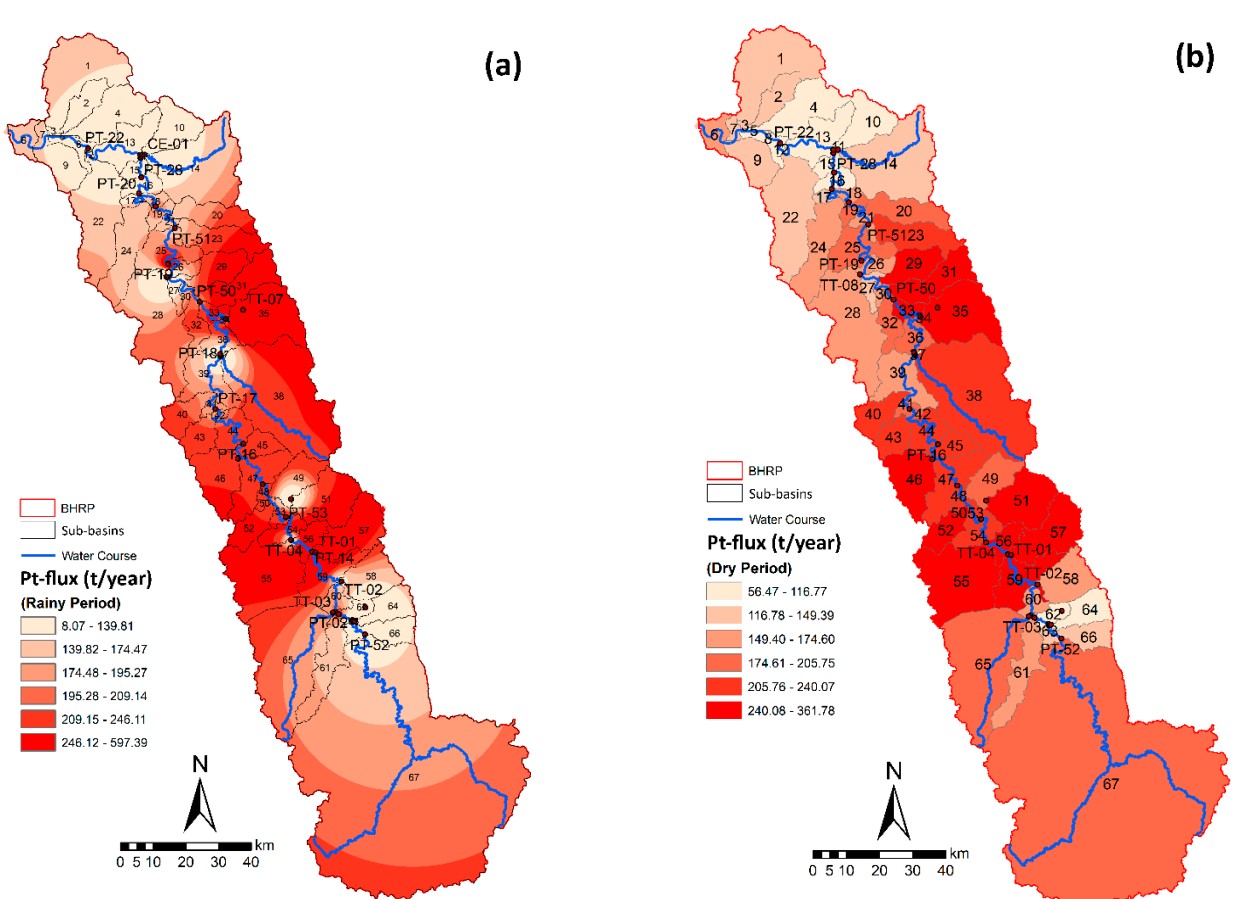

**Figure 13.** Pt fluxes in the Paraopeba River basin: (**a**) in the rainy period; and (**b**) in the dry season of 2019–2021 period.

### 3.4. Management Recommendations

Considering the results presented in the previous sections, it becomes evident that municipalities need to invest in basic sanitation infrastructures, effluent treatment plants, and facilities for an adequate disposal of solid waste, to reduce urban-related phosphorus pollution in the Paraopeba River basin. In addition, to effectively protect the riverine environment, it is paramount that farmers commit to adopt satisfactory soil and water management and conservation practices that are capable of containing erosive processes and ensuring a sustainable use of agricultural fertilizers, which implies investments in rural technical assistance, preservation and maintenance of riparian vegetation, and re-forestation of marginal areas, among others. In the sub-basins that currently exceed the legal Pt thresholds, producers could be asked to urgently reduce Pt levels in the water using best-management practices, such as performing periodic soil analysis and limiting phosphorus applications when indicated. Soil preparation practices could also be carried out to minimize erosion sub-basins that present a steep slope (20%), which should be a priority for these types of measures. They could also be the focus of action on riparian buffers, which should widen along the stream margins in order to prevent the entrance of Pt into the water network. As regards mining-related Pt, it is paramount to proceed with the monitoring of Paraopeba water and sediment quality, focused on metal and phosphorus concentrations, and to implement dredging in river stretches where sediments present anomalous concentrations of those elements, starting with the anomalous sector located between stations PT-13 and PT-17.

## 4. Conclusions

In 25 January of 2019, the Paraopeba River Basin (PRB) was affected by the rupture of B1 tailings dam of Córrego do Feijão mine (Vale, S.A. company, Rio de Janeiro, Brazil) located in the Ferro-Carvão stream and municipality of Brumadinho (Minas Gerais, Brazil). This study, based on box-plot and cluster analysis of total phosphorus concentrations (Pt) applied to 67 sub-basins of PRB delineated with the SWAT model, showed that this catastrophe increased rupture-related Pt to an average of 0.1 mg/L in a sector extending 8.8 to 155.3 km downstream from the B1 dam site, relative to lower values outside the segment. The sources contributing to these concentrations were presumably the tailings and soils removed from the Ferro-Carvão basin by the mud wave, which might have accumulated fertilizer-related phosphorus over the years. Therefore, the impact on phosphorus dynamics related to the B1 dam rupture is unquestionable. Other results from this study showed that the Pt in the studied period (2019–2021) were also determined by urban occupation (Betim town and areas from the Metropolitan region of Belo Horizonte), which raised Pt to a maximum value of 15.9 mg/L in 2019. Land use conversions involving the decrease of forest area and concomitant increase of urban, agriculture, and mining areas, also contributed to an increase of Pt, as did specific environmental settings. At this level of influence, the study showed a positive relationship between the cover of steep hillslopes (slope > 20%) with argisols and Pt increase. Taken altogether, the study highlighted the impact of multiple hazards on the Pt of a large multiple-use river basin, which is long-standing and has aggravated with the B1 dam rupture. The current diagnostic is preoccupied because Pt raised many times above the legal thresholds (e.g., >0.15 mg/L), mainly in the rainy season. It is therefore urgent that water authorities take action. The study recommended basic measures that are not in place yet, such as more ample wastewater treatment, but also rupture-related measures like the intensification of tailings dredging from the Paraopeba River channel.

Future studies will explore the full capacity SWAT through the development of a hydrological, sedimentological, and hydrochemical model skilled to predict Pt concentrations under a variety of scenarios, focused on the transport and dissemination of P-rich tailings along the Paraopeba River channel and their impact of river water quality.

**Supplementary Materials:** The following supporting information can be downloaded at: https://www.mdpi.com/article/10.3390/w14101572/s1, dataset of total phosphorus concentrations relative to the monitoring stations indicated in Figure 1c and to the January 2019–February 2021 period.

**Author Contributions:** Authors contributed equally to the manuscript. All authors have read and agreed to the published version of the manuscript.

**Funding:** This study was funded by the contract n°5500074952/5500074950/5500074953, signed between the Vale S.A. company (Rio de Janeiro, Brazil) and the following research institutions: Fundação de Apoio Universitário (Brazil); Universidade de Trás-os-Montes e Alto Douro (Portugal); and Fundação para o Desenvolvimento da Universidade Estadual Paulista Júlio de Mesquita Filho (Brazil). The author Renato Farias do Valle Junior received a productivity grant from the CNPq—Conselho Nacional de Desenvolvimento Científico e Tecnológico (Brazil). For the author integrated in the CITAB research centre (Portugal), this work was further supported by National Funds of FCT—Fundação para a Ciência e Tecnologia, under the project UIDB/04033/2020. The author integrated in the CITAB research centre is also integrated in the Inov4Agro—Institute for Innovation, Capacity Building, and Sustainability of Agri-food Production. The Inov4Agro is an associate laboratory composed of two R&D units (CITAB & GreenUPorto). For the author integrated in the CQVR (Portugal), the research was additionally supported by National Funds of FCT—Fundação para a Ciência e Tecnologia, under the projects UIDB/00616/2020 and UIDP/00616/2020.

**Institutional Review Board Statement:** Not applicable.

**Informed Consent Statement:** Not applicable.

**Data Availability Statement:** The dataset of total phosphorus concentrations used in the present study is provided as Supplementary Materials. Other data were compiled from public sources as indicated in Table 1.

**Conflicts of Interest:** The authors declare no conflict of interest. The funders had no role in the design of the study; in the collection, analyses, or interpretation of data; in the writing of the manuscript, or in the decision to publish the results.

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
