# Peer review of "Role of Mine Tailings in the Spatio-Temporal Distribution of Phosphorus in River Water: The Case of B1 Dam Break in Brumadinho"

_water, doi:10.3390/w14101572_

Round 1

Reviewer 1 Report

Highlight changes in yellow in a next revision, please. No track changes.

The manuscript has some similarity that, in English language… does seem so significant, but…, it relates content already published, starting from the title, that could be changed.

Assure to highlight novelty and innovation in order to differentiate the publications.

There are also parts of the texts with similarity and no cited reference: Introduction, for example.

This can be added as reference in the final list:

“To be used in this study, the BHRP was divided into compartments using the hydro- logical model Soil and Water Assessment Tool (SWAT) embedded in the ArcSWAT ex- tension (https://swat.tamu.edu/software/arcswat/).”

Also in tables:

“Table 1. Database of biogeophysical, chemical, climate and hydrologic parameters used in the pre- sent study.”

And several other cases

Assure that all images without references are original and details subcaptions by letter in grouped figures, after the main caption. The reader does not need to guess:

“Figure 1. Location of Paraopeba River basin (Minas Gerais, Brazil) in the parent São Francisco River 157 basin, and distribution of Minas Gerais state municipalities within and around the studied region. Identification of monitoring points used in this study, where phosphorus concentrations were as- sessed in the 2019 – 2021 period.”

And more…

“Figure 5. Box-plots of total phosphorus concentrations in stations located upstream of B1 tailings dam, with discrimination of season. Station PT-52 received drainage from sub-basin no. 67, station PT-01 from sub-basins no. 66 and 67, and station PT-47E from sub-basin no. 62.”

etc

Avoid abbreviations used in captions. Clarify captions to be self-explanatory…

“Figure 3. Elevation (a), slope (b), soil (c) and land use (d) in the BHRP.”

See that a caption just like this one is already found online. Check all cases

Avoid similar captions, either group or differentiate:

“Figure 10. Joint analysis of Pt concentrations and land uses based on a hierarchical cluster analysis applied to the 67 sub-basins. Symbols: pasture, forestry, lake, rocky, mining, agriculture, forest and urban, are land uses or occupations; Pt_ave, Pt_max and Pt_min are average, maximum and mini- mum total concentrations of phosphorus in water.”

“Figure 11. Joint analysis of Pt concentrations, terrain slope and soil types, based on a hierarchical cluster analysis applied to the 67 sub-basins. Symbols: latosol, cambisol, neosol and argisol, are soil types as indicated in Figure 3c; the slope classes are in keeping with the slope map of Figure 3b; Pt- ave, Pt-max and Pt-min are average, maximum and minimum total concentrations of phosphorus in water.”

Address italics to variables:

“Table 2. Spearman rank-order correlation coefficients between total phosphorus concentrations (Pt) and minimum (subscript min), maximum (max) and average (ave) stream discharge in the Paraopeba River basin in the rainy (Qr) and dry (Qd) seasons of 2019 – 2021 period. The coefficients highlighted in red are significant for p < 0.05.”

Add units to values, inside ()…, above

Add a heading to the first column…

Do not use red text, insulting in Europe, for example…

Check spacing: “(e.g., >0.15 mg/L)”

I liked to read the manuscript and it is clear what authors have made and achieved. It was a pleasure to read such an enlightening manuscript. I hope the comments above may assist the authors in improving minor apects.

Please clarify the countries, in each case. I would suggest using the translated English version of financing intuitions , when available…

“Funding: This study was funded by the contract”

Author Response

Response to Reviewer #1

We very much appreciate the time and effort made by the reviewer while going through our manuscript to revise it, allowing us to improve our work and produce a better publication. In the forthcoming paragraphs we indicate how we handled the reviewer’s comments and how we improved the manuscript in the sequel. We do it on a point-by-point basis, representing the reviewer comment in blue and Courier New font and our replies in black and Calibri font. In the revised manuscript the changes are represented as yellow shaded text.

Ad1) The manuscript has some similarity that, in English language… does seem so significant, but…, it relates content already published, starting from the title, that could be changed.

Answer: Many thanks for the comment. This study is a piece of a large scientific project (ENTIRE project, see the Funding section) dedicated to the rupture of B1 tailings dam that occurred in the Córrego do Feijão iron-ore mine located in the Ferro-Carvão stream, and affected the Brumadinho region and the Paraopeba River water quality from 2019 onwards. Eventually, this causes some context similarity among this and the previous publications, but we were cautious with the language across the manuscript as to reduce to a minimum the recycling (also known as self-plagiarism). We passed the final manuscript through the Plagiarism detector and found a 15% recycling, which is acceptable. A deeper look shows a single large portion of text copied from a published article, which is the funding declaration. If we remove that from the similarity check, the overall similarity reduces to 11%, which is absolutely normal. As regards the subject, this study is completely different from the previous works as this was focused on tailings-related phosphorus not addressed in the other works.

We agree that the title has also a similarity with the previous publications and for that reason we modified it in the revised version.

The original title was:

Modeling phosphorus in the Paraopeba River basin after the rupture of B1 tailings dam in Brumadinho (Minas Gerais, Brazil)

The new title is:

Role of mine tailings in the spatio-temporal distribution of phosphorus in river water: The case of B1 dam break in Brumadinho

Ad2) Assure to highlight novelty and innovation in order to differentiate the publications.

Many thanks for the pertinent suggestion. We added the following sentence to the revised manuscript

As, to our best knowledge, the assessment of tailings-related phosphorus is lacking in studies of river water quality involving dam breaks, the present work is novel in that regard.

Ad3) There are also parts of the texts with similarity and no cited reference: Introduction, for example.

Answer: As per our similarity check, we saw no large blocks of text in the Introduction, Materials and Methods, Results and Discussion or Conclusions sections. As we refer above, the funding declaration was copied from other publication, but this declaration cannot be changed. Besides that, only Table 1 (summary of data sources) contains more copied text because the databases used in this and other studies are mostly the same and hence the references to them cannot be changed (e.g., URL addresses). Some words, such as “Resampled” “Pixel size of 12.5 m x 12.5 m”, were not copied from anywhere, but were detected as so. Even though, in the revised version we changed the text to reduce this similarity.

Ad4 This can be added as reference in the final list: “To be used in this study, the BHRP was divided into compartments using the hydro- logical model Soil and Water Assessment Tool (SWAT) embedded in the ArcSWAT ex- tension (https://swat.tamu.edu/software/arcswat/).”

Answer: Thanks for the comment. Eventually, the reviewer is talking about the URL address. Yes, alternatively we could introduce a citation in the text as it was a paper or so, but we preferred to keep the URL written in the body of the text to simplify an immediate assess to the original webpage while a reader is passing through the paper. If the URLs are listed at the end, the reader will have to go back and fourth if He/She wants to assess the webpages.

Ad5) Also in tables: “Table 1. Database of biogeophysical, chemical, climate and hydrologic parameters used in the pre- sent study.” And several other cases

We answered to this in the reply to Ad3).

Ad6) Assure that all images without references are original and details subcaptions by letter in grouped figures, after the main caption. The reader does not need to guess:

“Figure 1. Location of Paraopeba River basin (Minas Gerais, Brazil) in the parent São Francisco River 157 basin, and distribution of Minas Gerais state municipalities within and around the studied region. Identification of monitoring points used in this study, where phosphorus concentrations were as- sessed in the 2019 – 2021 period.”

And more…

“Figure 5. Box-plots of total phosphorus concentrations in stations located upstream of B1 tailings dam, with discrimination of season. Station PT-52 received drainage from sub-basin no. 67, station PT-01 from sub-basins no. 66 and 67, and station PT-47E from sub-basin no. 62.”

etc

Answer: Thanks for the pertinent suggestion. We went through all captions and used letters to identify different panels, where applicable.

Ad7) Avoid abbreviations used in captions. Clarify captions to be self-explanatory…

“Figure 3. Elevation (a), slope (b), soil (c) and land use (d) in the BHRP.”

 Answer: Thanks for comment. However, as far as we could see the only abbreviation we used was the BHRP, which refers to the hydrographic basin of Paraopeba River and is used throughout the entire text many times. We believe that using this sole abbreviation will not make it difficult for the reader to guess what BHRP is.

Ad8) See that a caption just like this one is already found online. Check all cases

Answer: Ok. We changed the caption as to make it different to the on-line reference.

Ad9) Avoid similar captions, either group or differentiate:

“Figure 10. Joint analysis of Pt concentrations and land uses based on a hierarchical cluster analysis applied to the 67 sub-basins. Symbols: pasture, forestry, lake, rocky, mining, agriculture, forest and urban, are land uses or occupations; Pt_ave, Pt_max and Pt_min are average, maximum and mini- mum total concentrations of phosphorus in water.”

“Figure 11. Joint analysis of Pt concentrations, terrain slope and soil types, based on a hierarchical cluster analysis applied to the 67 sub-basins. Symbols: latosol, cambisol, neosol and argisol, are soil types as indicated in Figure 3c; the slope classes are in keeping with the slope map of Figure 3b; Pt- ave, Pt-max and Pt-min are average, maximum and minimum total concentrations of phosphorus in water.”

 Answer: The captions are similar but they refer to two runs of cluster analysis. However, the interveing variables are different the reason why we repeated the caption.

Ad10) Address italics to variables:

“Table 2. Spearman rank-order correlation coefficients between total phosphorus concentrations (Pt) and minimum (subscript min), maximum (max) and average (ave) stream discharge in the Paraopeba River basin in the rainy (Qr) and dry (Qd) seasons of 2019 – 2021 period. The coefficients highlighted in red are significant for p < 0.05.”

Add units to values, inside ()…, above

Add a heading to the first column…

Do not use red text, insulting in Europe, for example…

Check spacing: “(e.g., >0.15 mg/L)”

Answer: Done.

Add11) I liked to read the manuscript and it is clear what authors have made and achieved. It was a pleasure to read such an enlightening manuscript. I hope the comments above may assist the authors in improving minor apects.

Answer: We very much appreciate your kind words.

Add12) Please clarify the countries, in each case. I would suggest using the translated English version of financing intuitions , when available…

“Funding: This study was funded by the contract”

Answer: Done

Reviewer 2 Report

The objective of this work was to model P spatially in the Paraopeba River basin, namely in the main water course and 67 sub-basins, and temporally in the years of 2019, 2020 and 2021, after the rupture of B1 tailings dam of Vale, SA company in Brumadinho (Minas Gerais Brazil). The distribution of total phosphorus concentrations (Pt) in relation to environmental attritutes (terrain slope, soil class and land use) and stream flow, was assessed with the help of SWAT, the well-known Soil and Water Assessment Tool, coupled with box-plot and cluster analyses. The Pt were obtained from 33 sampling points monitored on a weekly basis. Mean values varied from 0.02 to 1.1 mg/L and maximum from 0.2 to 15.9 mg/L across the basin. The modeling results exposed an impact on the quality of Paraopeba River water in a stretch extending 8.8 to 155.3 km from the B1 dam, related with the rupture. In this sector, if the contribution from the rupture could be isolated from the other sources, the average Pt would be 0.1 mg/L. The highest Pt (15.9 mg/L) was directly proportional to the urban area of a sub-basin intersecting the limits of Betim town and Belo Horizonte Metropolitan Region. In general, urban sprawl as well as forest-agriculture and forest-mining conversions showed a close relationship with increased Pt, as did sub-basins with a predominance of argisols and an accentuated slope (>20%). There were various moments presenting Pt above legal thresholds (e.g., >0.15 mg/L), mainly in the rainy season.

The topic is interesting but there are some points to be addressed. The aim of the analysis should be further evidenced in the text. The methodological approach should be further explained for replication. The conclusions should be improved with the weaknesses of the analysis and the insights for future research. Finally, the manuscript should be English proofread because some sentences are not clear.

Author Response

Response to Reviewer #2

We very much appreciate the time and effort made by the reviewer while going through our manuscript to revise it, allowing us to improve our work and produce a better publication. In the forthcoming paragraphs we indicate how we handled the reviewer’s comments and how we improved the manuscript in the sequel. We do it on a point-by-point basis, representing the reviewer comment in blue and Courier New font and our replies in black and Calibri font. In the revised manuscript the changes are represented as yellow shaded text.

The objective of this work was to model P spatially in the Paraopeba River basin, namely in the main water course and 67 sub-basins, and temporally in the years of 2019, 2020 and 2021, after the rupture of B1 tailings dam of Vale, SA company in Brumadinho (Minas Gerais Brazil). The distribution of total phosphorus concentrations (Pt) in relation to environmental attritutes (terrain slope, soil class and land use) and stream flow, was assessed with the help of SWAT, the well-known Soil and Water Assessment Tool, coupled with box-plot and cluster analyses. The Pt were obtained from 33 sampling points monitored on a weekly basis. Mean values varied from 0.02 to 1.1 mg/L and maximum from 0.2 to 15.9 mg/L across the basin. The modeling results exposed an impact on the quality of Paraopeba River water in a stretch extending 8.8 to 155.3 km from the B1 dam, related with the rupture. In this sector, if the contribution from the rupture could be isolated from the other sources, the average Pt would be 0.1 mg/L. The highest Pt (15.9 mg/L) was directly proportional to the urban area of a sub-basin intersecting the limits of Betim town and Belo Horizonte Metropolitan Region. In general, urban sprawl as well as forest-agriculture and forest-mining conversions showed a close relationship with increased Pt, as did sub-basins with a predominance of argisols and an accentuated slope (>20%). There were various moments presenting Pt above legal thresholds (e.g., >0.15 mg/L), mainly in the rainy season.

The topic is interesting but there are some points to be addressed.

Ad1) The aim of the analysis should be further evidenced in the text.

Answer: Many thanks for the pertinent comment. we added the following sentences to the revised manuscript. “As specific objective, we aimed to verify if this catastrophic event raised significantly the concentrations of phosphorus in the river water and to what extend along the main channel, as well as if there were differences between the tailings-related contributions of phosphorus measured in the rainy and dry periods. As, to our best knowledge, the as-sessment of tailings-related phosphorus is lacking in studies of river water quality in-volving dam breaks, the present work is novel in that regard.”

Ad2) The methodological approach should be further explained for replication.

Answer: First, we segmented the Materials and Methods section in three subsections to clarify what is Study area description, dataset building and methods. Then, we improved the methods subsection by improving the following sentences:

A descriptive statistical analysis was performed in STATISTICA software (https://www.statistica.com/en/) to assess the central tendency, spread and trends in the Pt concentrations. The analysis was replicated for each monitoring station indicated in Figure 1 (panel c) to verify if the concentrations varied longitudinally along the main water course and, if yes, seek for a reason, namely the impact of B1 dam rupture. The results were visualized in the box-plots drawn for each station and built on the references of minimum and maximum values, first and third quartiles, median and outliers of Pt. The visualization in box-plots meant to detect seasonal variations in the Pt concentrations and check if the rupture of B1 tailings dam impacted the river water quality differently in the rainy and dry seasons, namely in the following periods: Rainy 2019 – January to March 2019, right after the B1 dam rupture; Dry 2019 – April to September 2019; Rainy 2020 – October 2019 to March 2020; Dry 2020 – April to September 2020; Rainy 2021 – October 2020 to March 2021.

Ad3) The conclusions should be improved with the weaknesses of the analysis and the insights for future research.

Answer: We improved the section informing about future studies that will extend the analysis through full implementation of SWAT in the assessment of P contamination in the Paraopeba River. “Future studies will explore the full capacity SWAT through the development of a hydrological, sedimentological and hydrochemical model skilled to predict Pt concen-trations under a variety of scenarios, focused on the transport and dissemination of P-rich tailings along the Paraopeba River channel and their impact of river water quality.”

Ad4) Finally, the manuscript should be English proofread because some sentences are not clear.

Answer: Done.
